# Wearable Drone Controller: Machine Learning-Based Hand Gesture Recognition and Vibrotactile Feedback

**DOI:** 10.3390/s23052666

**Published:** 2023-02-28

**Authors:** Ji-Won Lee, Kee-Ho Yu

**Affiliations:** 1KEPCO Research Institute, Daejeon 34056, Republic of Korea; 2Department of Aerospace Engineering, Jeonbuk National University, Jeonju 54896, Republic of Korea; 3Future Air Mobility Research Center, Jeonbuk National University, Jeonju 54896, Republic of Korea

**Keywords:** human–drone interface, wearable device, hand gesture recognition, machine learning, vibrotactile feedback

## Abstract

We proposed a wearable drone controller with hand gesture recognition and vibrotactile feedback. The intended hand motions of the user are sensed by an inertial measurement unit (IMU) placed on the back of the hand, and the signals are analyzed and classified using machine learning models. The recognized hand gestures control the drone, and the obstacle information in the heading direction of the drone is fed back to the user by activating the vibration motor attached to the wrist. Simulation experiments for drone operation were performed, and the participants’ subjective evaluations regarding the controller’s convenience and effectiveness were investigated. Finally, experiments with a real drone were conducted and discussed to validate the proposed controller.

## 1. Introduction

Nowadays, multicopter drones have been widely used because of their simple mechanism, control convenience, and hovering feature [1]. Drones are important in surveillance and reconnaissance, aerial photography and measurement, search and rescue missions, communication relay, and environmental monitoring [2,3]. To complete such applications and missions, highly sophisticated and dexterous drone control is required. Autonomous control has been partly used in their applications, as in waypoint following and programmed flight and mission, because of limited autonomy [4,5,6]. However, in the autonomous flight of drones, sometimes the autopilot is switched to manual control by a human operator according to the flight phase, such as landing, and in unexpected circumstances. The human role is necessary in the control loop when the system cannot fully reach an autonomous state.

Therefore, natural user interfaces for human operators have been studied extensively. An early study reported a novel user interface for manual control based on gestures, haptics, and PDA [7]. Moreover, multimodal natural user interfaces, such as speech, gestures, and vision for human-drone interaction, were introduced [8,9]. Recently, hand gesture-based interfaces and interactions using machine learning models have been proposed [10,11,12,13,14,15]. Hand gestures are a natural way to express human intent, and their effectiveness for applications of human–machine/computer interface/interaction has been reported in previous works. Some of the applications were focused on the control of drones based on deep learning models [16,17,18,19]. They used vision, optical sensors with infrared lights, and an inertial measurement unit (IMU) to capture the motion of the hand. The IMU attached to the user’s hand senses the motion of the hand robustly compared to conventional vision systems, which are easily affected by light conditions and require tedious calibrations. 

In contrast, to interact with a machine/computer, tactile stimulation has been adopted as a feedback means to humans for a long time [20,21,22]. Tactile stimulation is an additional channel to visual information for providing necessary information to humans. In particular, tactile feedback is important for visually impaired persons in terms of representing surrounding circumstances and the environment instead of having to rely on visual information [23,24]. Even for sighted people, tactile feedback helps improve their understanding of the environment when visual information is insufficient or blocked. Some previous work related to tactile and force feedback for drone control was reported [25,26,27]. Tsykunov et al. [25] studied a tactile pattern-based control glove for cluster flight of nano-drones. Cluster flights were intuitive and safe, in which the user’s position had to move continuously to guide the drone. Rognon et al. [26] addressed the flyjacket to control the drone with torso movements using IMU. The cable-driven haptic feedback system was mounted on the elbows of the user, who navigated the direction of waypoints. The flyjacket was demonstrated to control drones as much as joysticks and tablets, while the torso of the user was restricted. Duan et al. [27] developed a human–computer interaction system for improving realistic VR/AR. The tactile feedback device was used for estimating environments and objects. The gesture-based control was implemented with Leap Motion and Neural Network, which tend to be affected by light conditions.

We studied and implemented a wearable drone controller with hand gesture recognition and vibrotactile feedback, and the basic idea of the interface was presented as a preliminary version of this paper [28]. The hand gestures are sensed by the IMU, having robust features for motion acquisition compared to conventional vision placed on the back of the hand. The dominant parameters of the motion are extracted through sensitive analysis, and then machine learning-based classifications are performed. The classified and recognized hand gestures are commanded to the drone, and the distance to obstacles in the heading direction of the drone measured by the ultrasonic sensor is represented by activating the vibration motor attached to the user’s wrist. The IMU-based hand motion capture is relatively free in the distance between the drone and the user, and the drone does not need to make the pose to acquire the hand motion visually. Vibrotactile feedback is an effective way to obtain obstacle information around a drone, especially when the visual information is limited or blocked during operation of the drone.

The remainder of this paper is organized as follows. The system configuration of the wearable drone controller is presented in Section 2, and hand gesture recognition based on machine learning models is analyzed in Section 3. The simulation experiment with subjective evaluation of participants is performed in Section 4, and a real drone experiment for validation is presented in Section 5. Finally, Section 6 presents the conclusion.

## 2. System Configuration 

The controller developed here comprised vibrotactile feedback and hand gesture-based drone control. An ultrasonic sensor was mounted on the drone head to detect obstacles. A vibrator in the controller was stimulated according to the distance between the drone and its obstacles, as shown in Figure 1a. When the operator recognizes the vibration, the operator commands the gesture to avoid its obstacle (Figure 1b). The gestures were sensed by an IMU attached to the back of the hand. The gestures were categorized into two groups to control the drone appropriately. One of the control methods, called the direct mode, was defined to control drones directly for movements such as roll, pitch, and up/down. The cruise velocity of the drone was determined from the inclination of the IMU attached to the hand. Another control method, called the gesture mode, was used to control the drone with hand gestures more easily but may not perform quantitatively, as in the direct mode. The patterns of hand gestures used in the gesture mode were defined by imitating hand motions for helicopter guidance. Gesture pattern analysis and classification were conducted through machine learning to recognize these gestures, which are more sizable than the direct mode.

All sensors were connected to the Raspberry Pi 3b (Raspberry Pi Foundation, Cambridge, UK), as shown in Figure 2. The signal of the IMU was processed at 50 Hz and transmitted to the PC for gesture classification using MATLAB/Simulink. The classified gestures were also delivered to the AR drone control module of LabVIEW to command the drone. The ultrasonic sensor measured the distance, with the signal at 40 kHz. The distance information is transmitted wirelessly to the Raspberry Pi of the controller and converted to vibration intensity. The overall shape of the controller was fabricated using 3D printing.

### 2.1. Vibrotactile Feedback

In general, manual drone control is based on visual feedback. However, obstacles in the operating environment might interfere with visual feedback, which occasionally causes drone accidents. Hence, other sensory feedback, such as tactile and/or audio, is required to transfer obstacle information effectively.

In this study, vibrotactile feedback was adopted to represent obstacle information. An ultrasonic sensor (HC-SR04, SparkFun Electronics, Colorado, USA) was mounted on the drone head with a Raspberry Pi to operate the sensor so as to detect obstacles in front. The reliable detection region of the sensor was determined to be 0.03 to 2 m within ±15 °. The measured distance is transmitted to the Raspberry Pi of the controller to generate vibrotactile stimulation.

A coin-type vibration motor (MB-1004V, Motorbank Inc., Seoul, Korea) was used as the actuator for tactile stimulation. The motor’s size (diameter × height) was 9.0 × 3.4 mm, and its small size is suitable for wearable devices. The motor is attached to the wrist, which allows free hand motion for the intended gesture without any movement restriction. To effectively deliver vibration to the user, a tactile stimulator was designed as a bracelet, which comprised a spring and hemispherical plastic under the vibration motor, as shown in Figure 3.

Based on previous related studies [29,30,31], an amplitude of 100 µm and a frequency of 96.2 Hz were fixed as the stimulation conditions to satisfy the vibrotactile threshold of detection. The vibration intensity changes according to the distance between the drone and the obstacle. A preliminary experiment was conducted to determine appropriate stimulation conditions for the vibration motor. The vibration intensity was controlled by modulating the pulse width (PWM) applied to the vibration motor. The duty rate was changed by two levels with a step change because a minimum stimulation difference is required for human tactile perception [20]. At first, the duty rate of 50% is given as an initial vibration intensity when the measured distance to an obstacle is within 1.0 m, and then the duty rate is raised to 100% for strong vibration intensity when the measured distance is within 0.5 m and the drone is getting closer to the obstacle.

### 2.2. Geustre-Based Drone Control

Gestures facilitate the delivery of intentions, emotions, thoughts, and nonverbal communication. In particular, hand gestures have been widely used as effective and easy communication methods [32,33]. Hand gesture-based control provides intuitive and portable drone control. There have been studies on hand gesture recognition using vision-based sensors, such as RGB and depth cameras, and Kinect sensors [34,35]. However, these systems have limitations in terms of light, angle, and position of the environment. To overcome this problem, we propose a wearable sensor-based gesture recognition system. The wearable sensor-based method includes electromyography (EMG) and IMU, etc. [36,37]. We used an IMU (SEN-14001, SparkFun Electronics, Colorado, USA) which measures the 3-dimensional angle, angular velocity, and linear acceleration. 

As mentioned above, gestures were categorized into two groups corresponding to control purposes. The direct mode was defined to match drone movements such as roll, pitch, and up/down, as shown in Figure 4. The roll, pitch, and z-axis acceleration of the user’s hand were calculated using an inertial sensor and mapped to the drone’s posture and speed. The available hand motion range and its rate were considered to avoid the recognition of unintended hand gestures. A roll and pitch motion of less than ±30° and linear acceleration of less than ±10 m/s are disregarded for the stable recognition of hand gestures.

The gesture mode shown in Figure 5 is defined by imitating the hand signal used for helicopter guidance from the ground operator [38,39]. The gesture mode has difficulty controlling the drone quantitatively, as in the direct mode, but can be used more easily to control flight direction. These hand gestures are generated naturally with individual deviations; therefore, the patterns should be analyzed and classified for accurate recognition. This study adopted machine learning to learn and classify hand gestures.

## 3. Hand Gesture Recognition 

The signal processing scheme in hand gesture mode is illustrated in Figure 6.

In Figure 6, the signals of the hand movements are obtained from the inertial sensor of the controller. Subsequently, the key factors of gestures were determined using sensitivity analysis to reduce the calculation time of signal processing. Based on the analysis, key signals were segmented using sliding windows and filtered using a low-pass filter to eliminate the gravity component of the accelerometer. The features were extracted from the processed data, such as the mean, RMS, autocorrelation, power spectral density (PSD), and spectral peak. These features were input into the machine learning algorithm. Classification performance was evaluated based on accuracy, precision, recall, and the F-1 score.

### 3.1. Dataset

Large-scale motion data are required to classify hand gestures using a machine learning algorithm. Hence, the participants (male = 10, female = 2; right-hand dominant = 12; 26.6 ± 6, 24–30 years old) were asked to perform the motions of gesture mode (forward, backward, right, left, stop, up, and down). The signal was recorded at 50 Hz to obtain quaternion, acceleration, and angular velocity along the three axes. To prevent imbalanced data, the number of each gesture data was 3300, and the overall number of data was 23,100.

### 3.2. Sensitivity Analysis and Preprocess

A sensitivity analysis was conducted with reduced processing time to ensure that key parameters dominantly influence hand gestures. The results of the sensitivity analysis are shown in Figure 7. Based on the results, the acceleration elements of x, y, z, and the angular velocity of y and z are commonly influenced as key parameters of hand gestures. 

Raw signals were also processed to remove the gravity component forced on the accelerometer using a low-pass filter. These signals were segmented with fixed sliding windows of 2.56 s and a 50% overlap (128 readings/window) [40,41,42,43]. 

### 3.3. Feature Extraction 

Feature extraction is generally conducted to improve efficiency of the algorithm computation. The features are determined as time-domain and frequency-domain features [40,41,42,43,44,45]. We computed features such as the mean, RMS, autocorrelation, power spectral density (PSD), and spectral peak. The mean, RMS, and autocorrelation $r_{k}$ are included in the time-domain features. The characteristics are as follows:(1)$$mean=\frac{1}{N}\sum_{i=1}^{N}a_{i},$$
(2)$$RMS=\sqrt{\frac{1}{N}\left(a_{1}^{2}+a_{2}^{2}+\cdots +a_{N}^{2}\right)}$$
(3)$$r_{k}=\frac{\sum_{i=k+1}^{N}\left(a_{i}-\bar{a}\right)\left(a_{i-k}-\bar{a}\right)}{\sum_{i=1}^{N}{\left(a_{i}-\bar{a}\right)}^{2}}$$
where $a_{i}$ denotes the components of the acceleration and angular velocity, and N denotes the window length. Equation 3 computes autocorrelation, which is the correlation between $a_{i}$ and the delayed value $a_{i+k}$, where k = 0, ⋯, N, $\bar{a}$ denotes the mean of $a_{i}.$ PSD and spectral peaks were included in the frequency-domain features. PSD was calculated as the squared sum of the spectral coefficients normalized by sliding window length. PSD is described as follows [44,45]:(4)$$PSD=\frac{1}{N}\overset{N-1}{\underset{i=0}{\sum }}x_{i}^{2}+y_{i}^{2}$$
with $x_{i}=z_{i}\;\cos\left(\frac{2\pi fi}{N}\right)$ and $y_{i}=z_{i}\;\sin\left(\frac{2\pi fi}{N}\right)$, where z denotes the discrete data for the frequency spectrum, and f represents the Fourier coefficient in the frequency domain. The spectral peak was calculated as the height and position of PSD [44,45].

### 3.4. Classification Model

To employ an appropriate classification algorithm, we compared the classification results of the ensemble, SVM, KNN, naive Bayes, and trees. The classifiers were used in MATLAB, and the training and test data were used in 9 to 1.

The performance of the learned model was evaluated based on the following expressions:(5)$$Accuracy=\frac{TP+TN}{TP+TN+FP+FN}$$
(6)$$Precision=\frac{TP}{TP+FP}$$
(7)$$Recall=\frac{TP}{TP+FN}$$
(8)$$F1-score=2*\frac{recall*precision}{recall+precision}$$
where TP is a true positive, TN is a true negative, FP is a false positive, and FN is a false negative. Figure 8a illustrates the comparison results of performance. Figure 8b shows the detailed results of each classifier’s accuracy. According to Table 1, the ensemble method exhibited adequate performance in terms of classification accuracy.

## 4. Simulation Experiment

### 4.1. Gesture-Based Control

#### 4.1.1. Simulation Setup

To assess the effectiveness of the proposed gesture control, the participants (male = 10, female = 2; right-hand dominant = 12; 26.6 ± 6, 24–30 years old) performed a virtual flight simulation with hand gestures while equipped with a wearable device. The participants did not have experience flying drones or were beginners. A schematic of the simulation mission is shown in Figure 9. The participants executed the appropriate hand gesture to maneuver the drone through the 12 waypoints described by the passing windows. These waypoints were placed according to the yellow guideline to pass, and one waypoint was required to adjust the flight altitude of the drone. The position of the drone was calculated at 50 Hz through quadrotor dynamics from MATLAB/Simulink. The simulation experiment comprised three preliminary sessions and the main experiment with a given mission. All participants were given sufficient time to complete a flight mission with direct and gesture modes.

Average trial duration, gesture accuracy, and number of gesture repetitions were calculated to evaluate the learning effect and adaptability of gesture-based drone control. Average trial duration, meaning the required average time for task completion, was used to evaluate the learning effect in the subjects. The average gesture accuracy was used to evaluate the performance of the classification of hand gestures. The average number of gesture repetitions, counting the frequency of motion changes, was used to evaluate adaptability. Participants were also asked to respond to a questionnaire regarding convenience of the control operation and physical discomfort after the experiment. Questions were answered in the form of a 7-point Likert scale.

#### 4.1.2. Results

The results of the first and last trials were analyzed to investigate the controller’s efficiency and the learning effect. Figure 10 and Figure 11 indicate the drone trajectory and the frequency of motion changes during the first and last attempts using direct mode control. Compared with the first trial, the trajectory stabilized in the last trial. In addition, the frequency of motion changes decreased because of the adaptation of direct mode control. Average accuracy also increased as the experiment progressed, and there was a larger reduction in trial duration and repetition, as shown in Figure 12.

Similarly, Figure 13 and Figure 14 show drone trajectory and frequency of motion changes in the first and last trials with gesture mode control. In the first trial, the drone trajectory was unstable, especially when the user executed the gesture in the left direction. The frequency of motion changes also decreased compared with the first trial. Across all trials, average accuracy tended to increase, as did a larger reduction in trial duration and repetition, as shown in Figure 15. These results indicate that gesture-based control has learning effects and adaptability.

### 4.2. Vibrotactile Feedback

#### 4.2.1. Simulation Setup

An obstacle avoidance simulation experiment was conducted to demonstrate the performance of vibrotactile feedback. The participants, who were the same as mentioned above, were asked to avoid obstacles distributed in front of the drone using gesture control. All the participants performed the simulation experiments three times with a fixed field of view, as shown in Figure 16a. In the first trial, participants controlled the drone without vibrotactile conditions. In the other trials, the drone was controlled using vibrotactile feedback. To assess the performance of the vibrotactile stimulation, the evaluator measured the collision status through the top view of the flight environment, as shown in Figure 16b.

#### 4.2.2. Result

Without vibrotactile feedback, the participants could not manipulate the drone appropriately because of a lack of visual feedback, as shown in Figure 16a. The drone successfully avoided all obstacles with vibrotactile feedback, as illustrated in Figure 17.

The results of the vibrotactile feedback simulation are presented in Table 2. All the participants experienced crashes with obstacles at least once. When the vibrotactile stimulator was not used, the participants did not recognize the obstacles well with limited visibility. Therefore, the drone crashed an average of 2.2 times. However, with vibrotactile feedback, each participant detected obstacles via vibrotactile stimulation in the wrist and effectively commanded the drone to avoid obstacles.

### 4.3. Subjective Evaluation of Participants

Subjective evaluations were conducted to assess the user-friendliness and effectiveness of the controller after the experiment. Participants were asked to respond to the experience of use based on a 7-point Likert scale. The participants gave assessment points for the user-friendliness and effectiveness of the direct and gesture modes. Similarly, participants were asked to respond to the control with and without vibrotactile feedback. Table 3 shows the subjective evaluation of gesture-based drone control. The participants agreed to the convenience and effectiveness of the controller by more than 6 points and did not agree on discomfort and fatigue by less than 3 points. There are no significant differences between the direct and gesture modes, confirming that the proposed controller is easy to use and effective.

Table 4 shows the subjective evaluations of the vibrotactile feedback. All the participants responded that there were substantial differences between the control with vibrotactile feedback and the control relying on only limited visual information. They expected that vibrotactile feedback would be helpful for drone control in the field.

## 5. Experimental Validation

### 5.1. Gesture-Based Drone Control

#### 5.1.1. Setup

The implemented controller was tested to validate its performance. The flight test was conducted using a quadrotor (AR.Drone 2.0, Parrot, Paris, France). In the mission scenario, the user flew the drone through four gates using gesture control, as shown in Figure 18. The drone cruised at 0.4 m/s and hovered at a height of about 0.8 m. The user was positioned within gray dotted lines (Figure 18a) to ensure visibility for drone control. Three gates were placed on the guideline, and one was installed on the table to control up/down movements. An appropriate gesture was assigned for each trajectory, and the user’s gesture was recognized and compared with the assigned gesture. To evaluate the performance, gesture accuracy was evaluated using drone states, classified gestures, and aerial videos during the experiment.

#### 5.1.2. Result

The entire recognition rate in the direct mode is shown in Figure 19a. The average accuracy of the direct mode was 96.4%, and the mission duration was 103 s. The classification rate in the gesture mode condition is shown in Figure 19b. The average accuracy of the gesture mode was 98.2%, and the mission duration was 119 s.

### 5.2. Vibrotactile Feedback

#### 5.2.1. Setup

Vibrotactile feedback was tested for efficiency in a real environment, as shown in Figure 20. The user controls the drone using gesture mode through gates to avoid obstacles. To demonstrate the effectiveness of vibrotactile feedback, the user’s visibility was limited by the blinding of the panel, which blocked part of the obstacle and the gate to pass. The first trial was performed without a vibrotactile condition. The second trial was conducted with vibrotactile feedback, representing the distance between the drone and the obstacle.

#### 5.2.2. Result

The drone failed to avoid collisions without vibrotactile feedback, as shown in Figure 21a. The user cannot estimate the distance between the drone and the obstacle owing to limited visual information. However, collision avoidance was achieved successfully with vibrotactile feedback, as shown in Figure 21b. In addition, the distance to the obstacle in front of the drone was inferred through the intensity of the vibrotactile stimulation, and the mission was carried out by maintaining a certain distance from the obstacle.

### 5.3. Discussions

The presented drone controller exhibited user-friendly and intuitive features, which can be used more easily by an inexperienced user for drone maneuvering. Through simulation experiments on drones, an effective interface for drone control was revealed by estimating control accuracy and mission duration. An experiment on a real drone was carried out to validate the effectiveness of the proposed controller. Indeed, vibrotactile feedback was helpful in detecting obstacles compared to using only visual information. The participants also felt more comfortable with and interested in the proposed controller.

In a further study, the addition of yawing control of the drone was considered to implement a supplementary sensor to measure the hand motion’s orientation accurately. Furthermore, implementing additional ultrasonic sensors on a drone enables the omnidirectional distance measurement of obstacles. The aforementioned approaches are expected to complete a wearable drone control system with a natural user interface.

## 6. Conclusions

This study proposed a wearable drone controller with hand gesture recognition and vibrotactile feedback. The hand motions for drone control were sensed by an IMU attached to the back of the hand. The measured motions were classified based on machine learning using the ensemble method with a classification accuracy of 97.9%. Furthermore, the distance to obstacles in the heading direction of the drone was fed back to the user stimulating the vibration motor attached to the wrist, known as vibrotactile feedback. In the simulation experiment by the participant group, the hand gesture control showed good performance, and the vibrotactile feedback helped the user be aware of the operation environment of the drone, especially when limited visual information was available. A subjective evaluation of the participants was performed to assess the convenience and effectiveness of the proposed controller. Finally, an experiment with a real drone was conducted, confirming that the proposed controller could be applicable for drone operation as a natural interface.

## Figures and Tables

**Figure 1 sensors-23-02666-f001:**
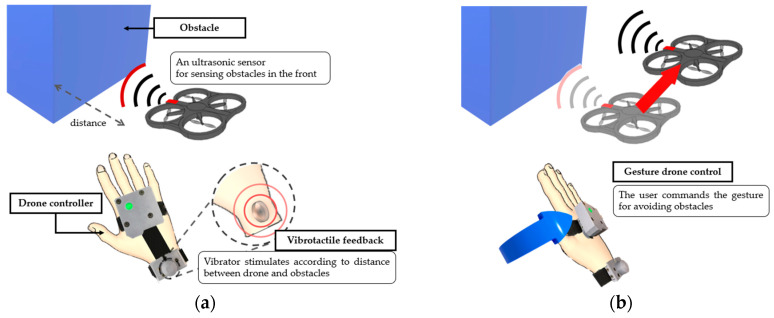
Concept of the proposed controller with vibrotactile feedback and gesture-based drone control. (**a**) Obstacle is detected in front of the drone, and the vibrator is stimulated. (**b**) Gesture of right direction is performed to avoid obstacle using gesture control.

**Figure 2 sensors-23-02666-f002:**
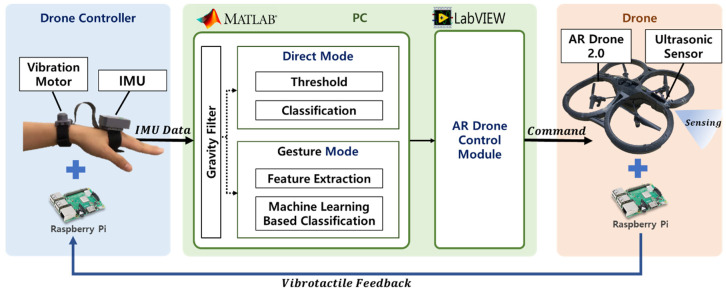
System configuration of wearable drone interface comprises a drone controller, hand gesture recognition and drone control module in PC, and drone.

**Figure 3 sensors-23-02666-f003:**
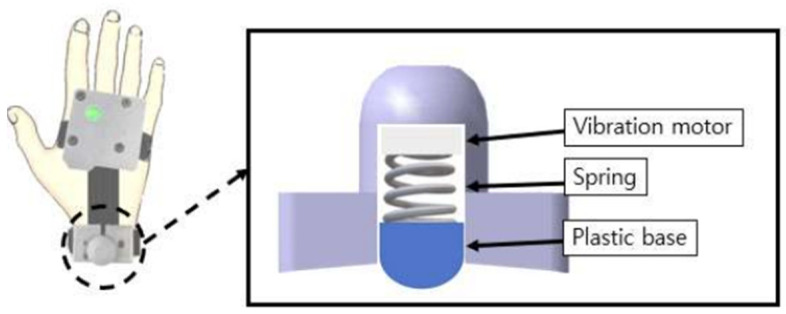
Structure of vibrotactile stimulator.

**Figure 4 sensors-23-02666-f004:**
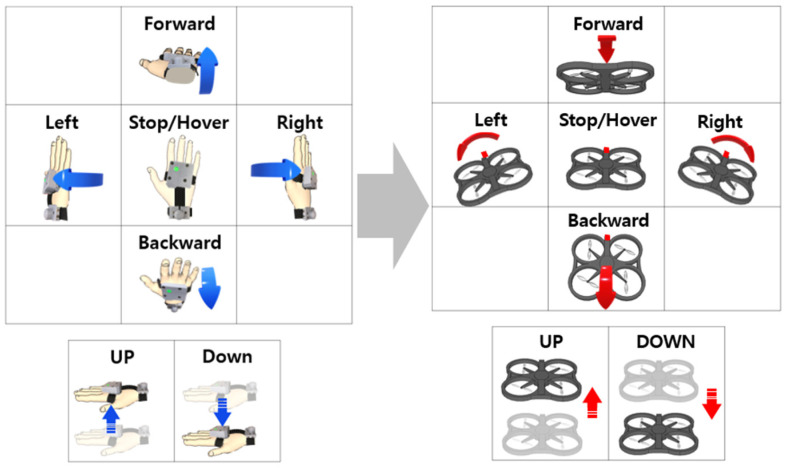
Definition of direct mode. It can directly control the drone’s posture and speed according to inclination and up/down of hand.

**Figure 5 sensors-23-02666-f005:**
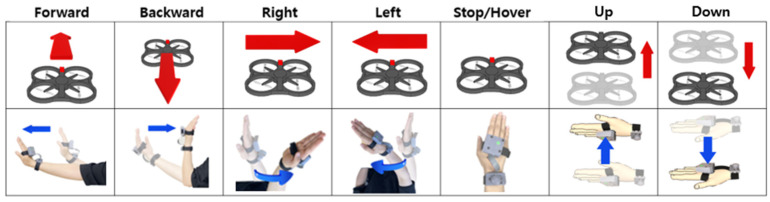
Definition of gesture mode. Gesture mode can control the direction of drone with natural hand motion.

**Figure 6 sensors-23-02666-f006:**
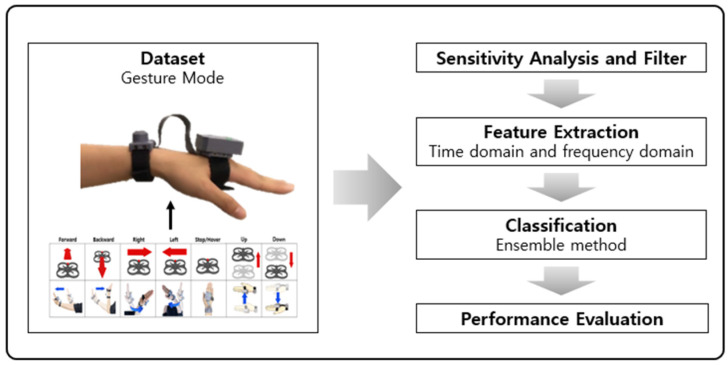
Scheme of signal processing for hand gesture recognition.

**Figure 7 sensors-23-02666-f007:**
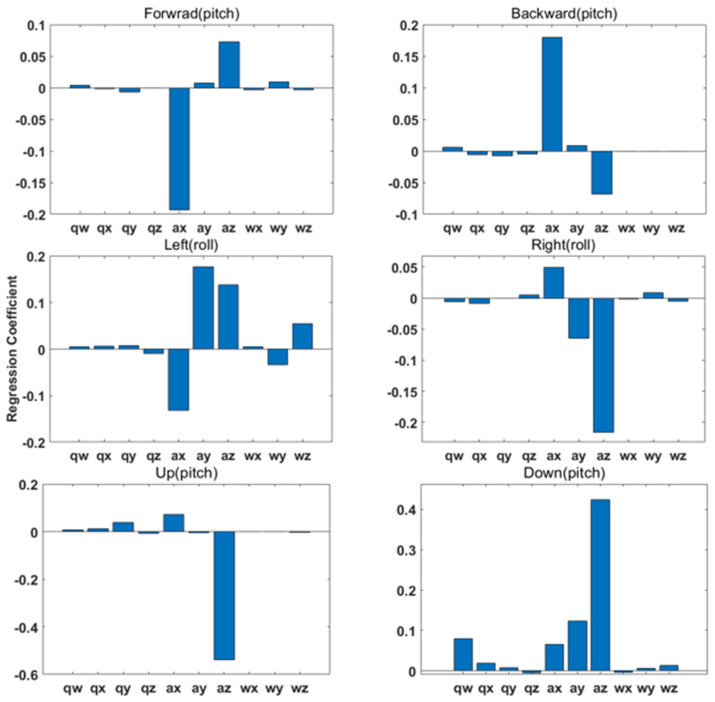
Result of sensitivity analysis on gesture mode movements. In particular, gestures such as forward, backward, up, and down were found to perform numerous pitch motions. Left and right gestures were known to perform roll actions primarily. According to the results, acceleration of x, y, z and angular velocity of y and z were determined to be key parameters in common.

**Figure 8 sensors-23-02666-f008:**
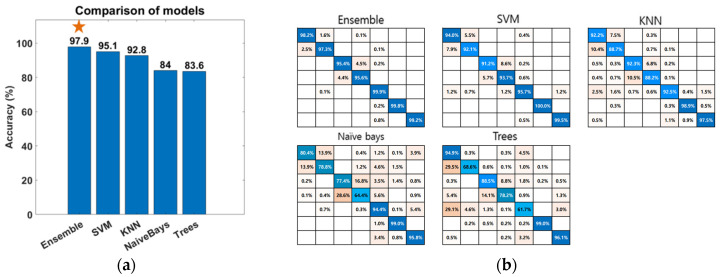
Result of gesture mode classification: (**a**) Comparison accuracy of each classification model. (**b**) Confusion matrix of ensemble, SVM, KNN, naive Bayes, and trees. From top left, the hand motions of forward, backward, right, left, stop (hover), up, and down are presented in order.

**Figure 9 sensors-23-02666-f009:**
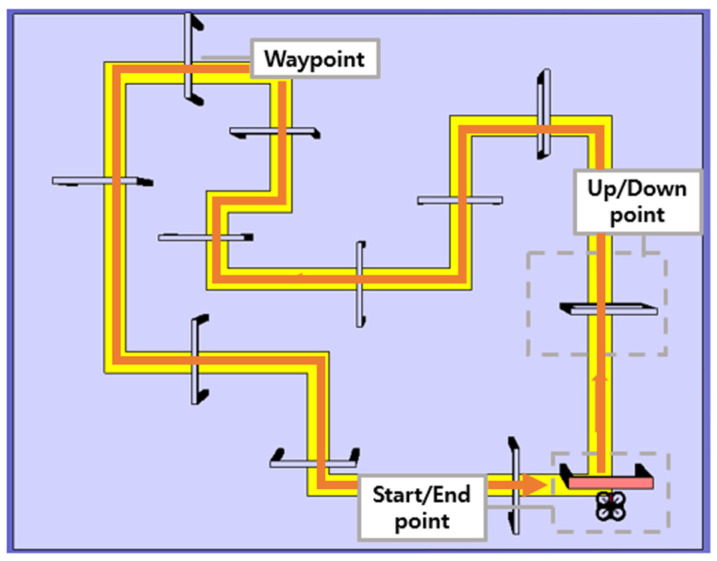
Mission schematic for drone flight simulation with gesture-based drone control.

**Figure 10 sensors-23-02666-f010:**
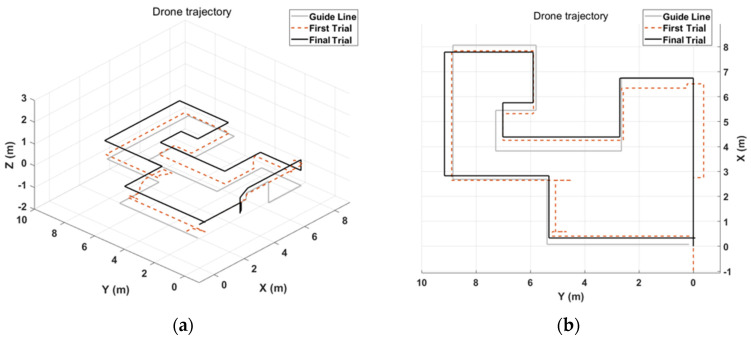
Example of drone flight trajectory with direct mode. Gray line is guideline, dotted line is trajectory of first trial, and black line is trajectory of last trial. Comparing the trajectory between first and last trials, the last one is a more stable shape. (**a**) Isometric view of drone trajectory can be found by adjusting drone altitude. (**b**) Top view of drone trajectory can be determined by behavior of drone according to the user on the x-y dimension.

**Figure 11 sensors-23-02666-f011:**
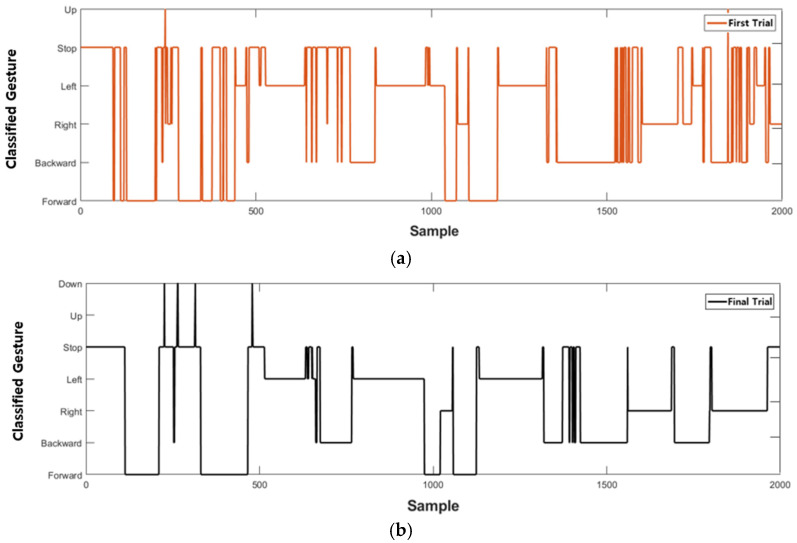
Frequency of gesture changes during trial duration with direct mode. (**a**) Classified gesture changes at first trial. (**b**) Classified gesture changes at last trial, which shows less change than first trial.

**Figure 12 sensors-23-02666-f012:**
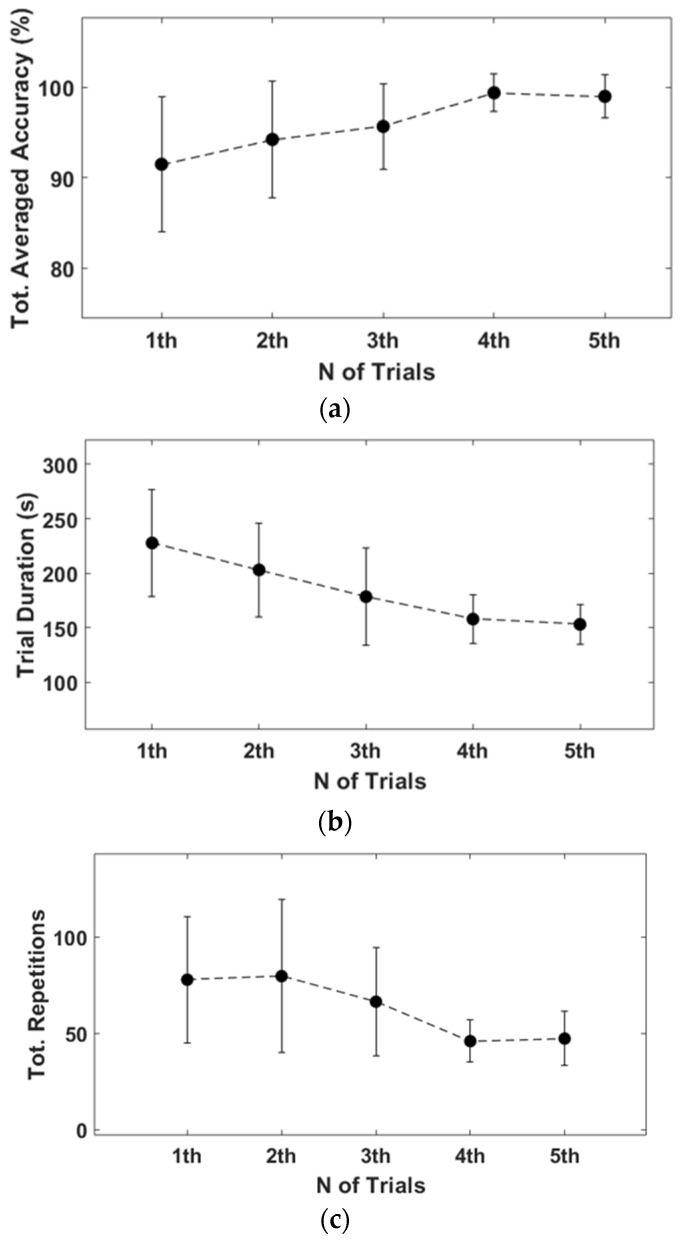
Results of direct mode simulation (mean ± SD). (**a**) Total average accuracy of each trial. First trial is 91.5 ±7.4%, and last trial is 99 ± 2.4%. (**b**) Average trial duration. First trial is 227.7 ± 48.7 s, and last trial is 153.0 ± 18.1 s. (**c**) Total repetitions of gestures. First trial is 77.8 ± 32.8, and last trial is 47.2 ± 14.0.

**Figure 13 sensors-23-02666-f013:**
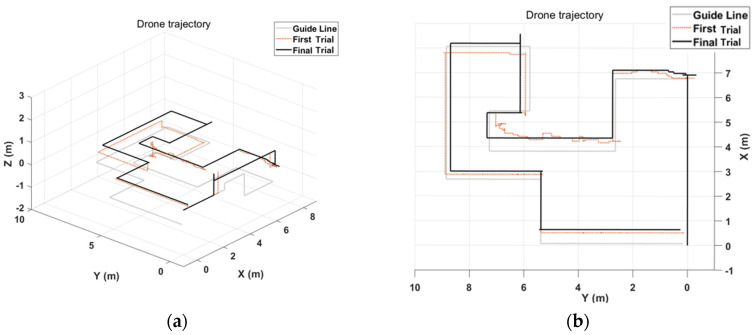
Example of drone flight trajectory with gesture mode. Gray line is guideline, dotted line represents drone trajectory of first trial, and black line is last trial. (**a**) Isometric view can be found altitude control using gestures. (**b**) Top view of drone trajectory can be figured out drone movements on x-y dimension. While first trial is low-accuracy and the trajectory is unstable, last trial is high-accuracy and confirms that the trajectory appeared similar to the guideline.

**Figure 14 sensors-23-02666-f014:**
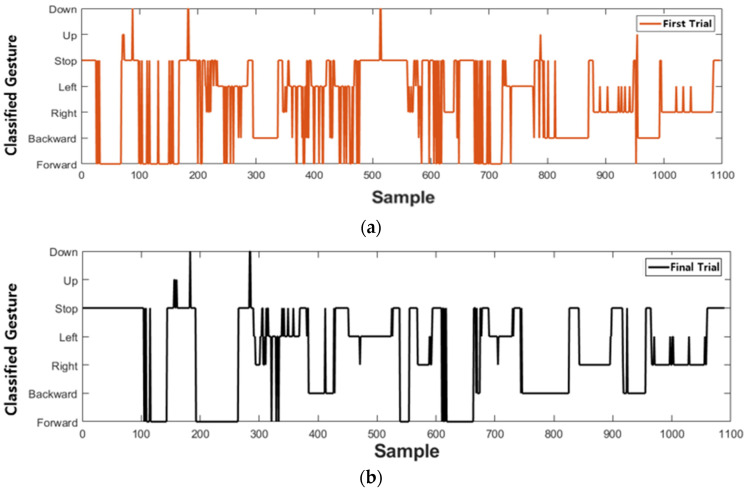
Frequency of gesture changes during trial duration with gesture mode. (**a**) Classified gesture changes at first mission. (**b**) Classified gesture changes at last trial, which shows less change than first trial.

**Figure 15 sensors-23-02666-f015:**
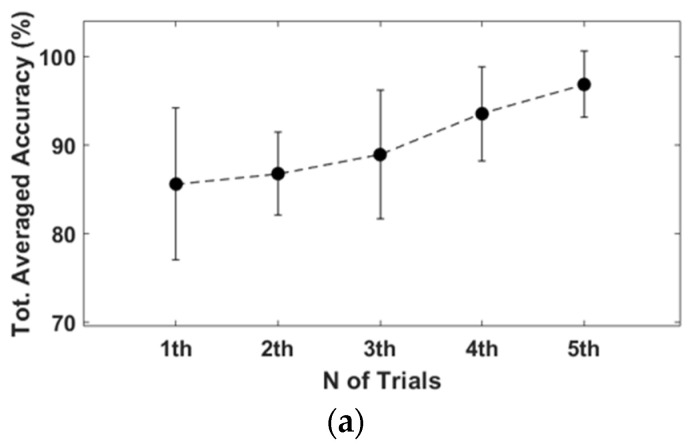
Results of gesture mode simulation (mean ± SD). (**a**) Average accuracy of first trial was 85.9 ± 8.4%, and final trial was 96.8 ± 3.9%. (**b**) First trial duration was 264 ± 73.3 s, and last duration was 155.7 ± 28.9 s, which shows decreased duration. (**c**) Difference of total repetitions between first and last trial is 138.9 ± 59.1.

**Figure 16 sensors-23-02666-f016:**
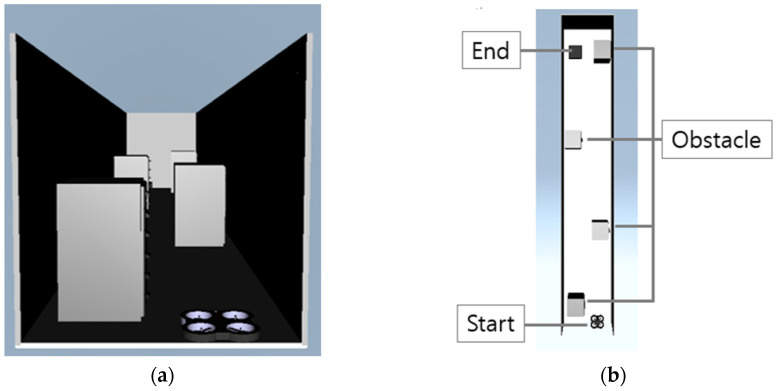
Mission schematic for vibrotactile feedback simulation experiment. (**a**) Fixed view from the participants who were asked to control the drone flight avoiding obstacles. (**b**) Top view of drone flight environment to evaluate the control performance with vibrotactile feedback.

**Figure 17 sensors-23-02666-f017:**
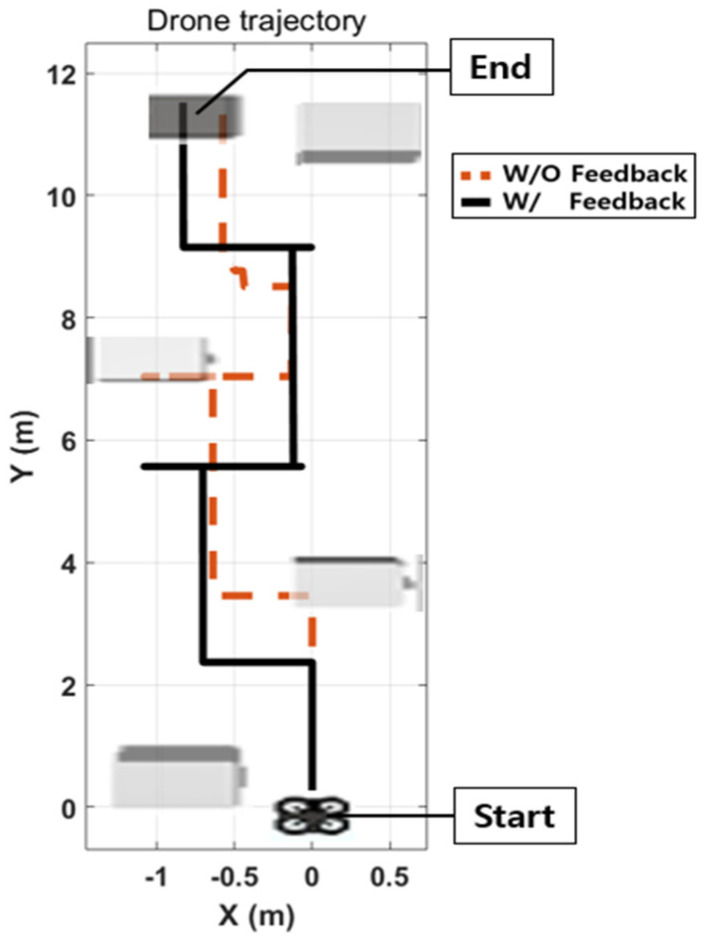
Example of drone flight trajectories with and without vibrotactile conditions. When the vibrotactile feedback did not work, the drone crashed into boxes because of lack of visual feedback (dashed line). With vibrotactile feedback, the drone achieved the mission of avoiding obstacles (black line).

**Figure 18 sensors-23-02666-f018:**
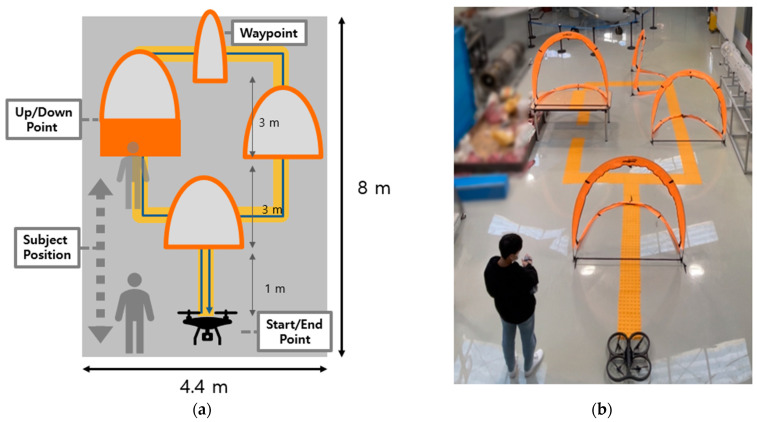
Experiment scenario with gesture-based drone control. (**a**) Schematic diagram for experiment. The yellow guideline was installed for reference of drone trajectory. The user walked within gray dotted lines to secure the view. (**b**) Experiment environment based on schematic was set indoors for safe operation.

**Figure 19 sensors-23-02666-f019:**
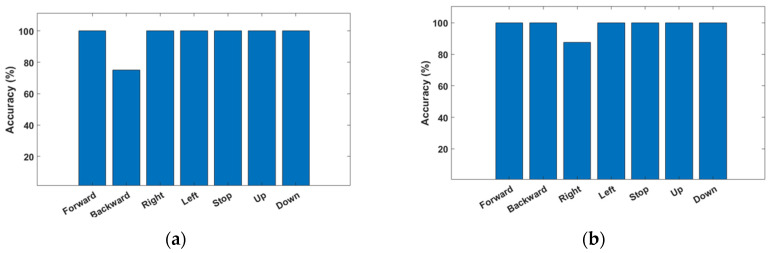
Accuracy of real drone control. (**a**) Average accuracy of direct mode is about 96%, which is about 2.6% lower than the simulation. (**b**) Average accuracy of gesture mode is about 98%, which is about 1.4% higher than the simulation.

**Figure 20 sensors-23-02666-f020:**
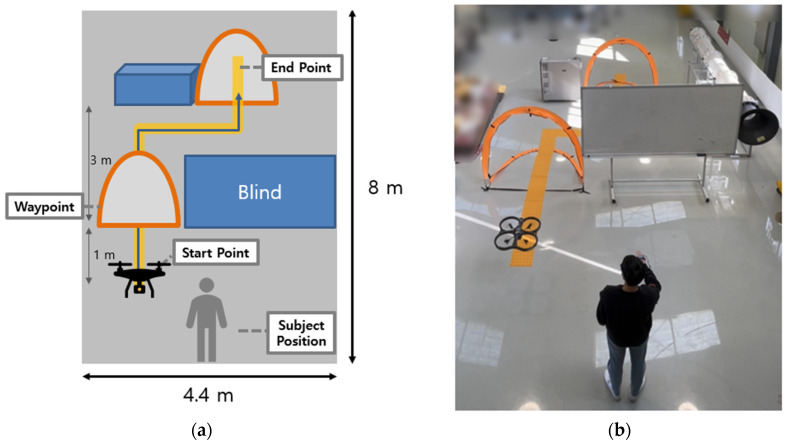
Flight experiment for vibrotactile feedback with gesture mode. (**a**) Schematic diagram comprising visual obstruction, drone, passing gates, obstacle, and guideline. (**b**) Based on the scheme, flight experiment was set up.

**Figure 21 sensors-23-02666-f021:**
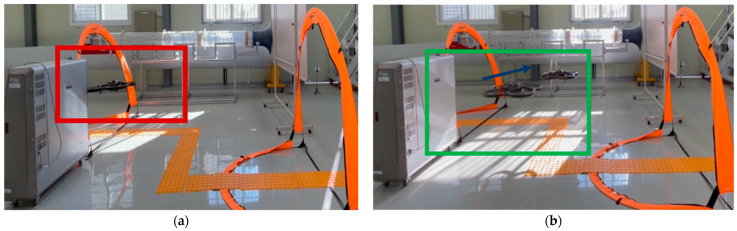
Result of vibrotactile feedback. (**a**) Drone could not avoid the collision in front. Owing to the lack of visual information for the obstacle, the mission failed. (**b**) Drone flew successfully through gates, and distance to the obstacle was maintained using vibrotactile feedback.

**Table 1 sensors-23-02666-t001:** Comparison result of classifiers (%).

	Ensemble	SVM	KNN	Naive Bayes	Trees
accuracy	97.9	95.1	92.8	84.0	83.6
precision	97.9	95.1	93.0	84.1	83.8
recall	97.9	95.1	92.9	84.1	83.8
F1-score	97.9	95.1	92.9	84.1	83.8

**Table 2 sensors-23-02666-t002:** Results of obstacle avoidance using vibrotactile feedback. Without vibrotactile feedback, the drone failed to avoid obstacles; meanwhile, collision avoidance was completed successfully with vibrotactile feedback.

Participant No.	First Trial (without Feedback)	Second Trial (with Feedback)	Third Trial (with Feedback)
Participant 1	2/4 *	4/4	4/4
Participant 2	1/4	4/4	4/4
Participant 3	1/4	4/4	4/4
Participant 4	2/4	4/4	4/4
Participant 5	1/4	4/4	4/4
Participant 6	2/4	4/4	4/4
Participant 7	2/4	4/4	4/4
Participant 8	1/4	4/4	4/4
Participant 9	2/4	4/4	4/4
Participant 10	2/4	4/4	4/4
Participant 11	2/4	4/4	4/4
Participant 12	2/4	4/4	4/4

* Successful/avoidance trials.

**Table 3 sensors-23-02666-t003:** Results of subjective evaluation by participants in gesture-based drone control.

Question	Direct Mode	Gesture Mode
1. The proposed gesture was natural for me.	6.4 ± 0.6	6.0 ± 0.7
2. I felt physical discomfort while controlling.	1.6 ± 0.6	2.0 ± 0.9
3. My hand and arm were tired while controlling.	2.0 ± 0.6	2.4 ± 0.8
4. The proposed gesture was user-friendly.	6.5 ± 0.9	6.3 ± 1.4
5. I felt the convenience of controlling a drone with one hand.	6.5 ± 0.6	6.6 ± 0.5
6. It was interesting to fly a drone with a gesture.	6.5 ±1.0	6.9 ± 0.3

**Table 4 sensors-23-02666-t004:** Results of subjective evaluation by participants in vibrotactile feedback.

Question	Mean ± SD
1. The vibrotactile feedback was helpful for obstacle avoidance.	6.9 ± 0.3
2. The vibration intensity was appropriate.	6.6 ± 0.6
3. My hand and wrist were tired by vibrotactile feedback.	1.4 ± 0.8
4. The obstacle avoidance was difficult without the vibrotactile feedback condition.	6.5 ± 0.7
5. If I flew a drone in real life, vibrotactile feedback would be helpful.	6.5 ± 0.3

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
