# Peer review of "Wearable Drone Controller: Machine Learning-Based Hand Gesture Recognition and Vibrotactile Feedback"

_sensors, 2023, doi:10.3390/s23052666_

Round 1

Reviewer 1 Report

Introduction:

The paper addresses the issue of controlling an aerial drone with a wristband equipped with inertial sensor. The wristband provides feedback to the user  about obstacles distances thanks to acoustic sensors equipping the drone.

The paper is organised as follows:

Section 1 introduces the context of the study and the scope of the research activities. Part 2 is devoted to the description of the wristband wearable. Hand gestures recognition based on machine learning models is studied and explored in section 3. The simulation experiment on hand gesture capture  conducted with participants is performed in Section 4. Finally, section 5 describes experiment conducted with the overall system architecture.

General comment:

The paper is well organised and soundness but suffers from a lack of novelty. In fact, the issue of controlling aerial drone with wearables is not a new topic. There are a number of scientific publications addressing this topic. Furthermore, many commercial products are now available in the market. In addition, I’m relatively sceptical about using wearables for controlling drones. In fact, drones need nowadays to be controlled precisely with accurate movements under various applications fields including inspection, SLAM, military intervention. Having in mind that MEMS IMUs are still suffering from data imprecision (bias), controlling drone with wearables cannot be as good as joysticks or pads controllers. The authors have probably to either motivate or justify better the benefit of using wearables comparing to conventional solutions.

  Additional comment:

-  It’s not clear how does the machine learning approach is providing an added benefit for classifying hand gestures. In fact, the sensitivity analysis provided in section 3 shows a good agreement between the hand gesture and data. For instance, the forward and backward movements indicate that acceleration ax is the most represented and relevant as controlling data comparing to others acceleration’s directions.  I’m wondering for the sake of clarity why these data are not just used simply to control the drone.  

-  Unless I’m mistaken, the system used to track the drone position (gold standard/reference) is not described in the paper. Figure 10 shows measurement of the position of the drone thanks to several trials. Please introduce the technical features of the tracking system.

- Widening the number of participants would improve significally the soundness of the experiments and enables to identify the effect of the learning curve.    

Author Response

Statement of Changes

Submission number: sensors-2183113

Wearable Drone Controller: Machine Learning-Based Hand Gesture Recognition and Vibrotactile Feedback

Ji-Won Lee and Kee-Ho YU

Dear Reviewers,

Thank you for your review, and we really appreciate the effort of the review and the comments from you. Your advice was invaluable to improve the quality of the paper and provided insight to carry out the further work.

According to the comments of the review, the paper was revised carefully throughout the previously submitted manuscript. The added parts of the paper are highlighted in yellow color and the reply to your comments/suggestions are addressed below each comment, respectively.

List of Changes

  1. (p2) Added the description of the related work and limitations.
  2. (p2) Added the reason of IMU instead of vision.
  3. (p8) Added the experience of participants about drone control.
  4. (p8) Added the technical feature of tracking system in the simulation.

Review #1

The paper addresses the issue of controlling an aerial drone with a wristband equipped with inertial sensor. The wristband provides feedback to the user about obstacles distances thanks to acoustic sensors equipping the drone.

The paper is organized as follows:

Section 1 introduces the context of the study and the scope of the research activities. Part 2 is devoted to the description of the wristband wearable. Hand gestures recognition based on machine learning models is studied and explored in section 3. The simulation experiment on hand gesture capture conducted with participants is performed in Section 4. Finally, section 5 describes experiment conducted with the overall system architecture.

  1. The paper is well organized and soundness but suffers from a lack of novelty. In fact, the issue of controlling aerial drone with wearables is not a new topic. There are a number of scientific publications addressing this topic. Furthermore, many commercial products are now available in the market. In addition, I’m relatively sceptical about using wearables for controlling drones. In fact, drones need nowadays to be controlled precisely with accurate movements under various applications fields including inspection, SLAM, military intervention. Having in mind that MEMS IMUs are still suffering from data imprecision (bias), controlling drone with wearables cannot be as good as joysticks or pads controllers. The authors have probably to either motivate or justify better the benefit of using wearables comparing to conventional solutions.

Thank you for your comments. Although autonomous flight has been developed, manual control by human operators is necessary, especially under unexpected circumstances and/or accomplishing complex mission. As opposed to joysticks, wearable controllers are intuitive, easy-to-use, and can save the time to learn to operate drone [26]. We agree that wearable controller using IMU can cause data imprecision, but it has been improved by filtering and preprocessing in this study. This was stated in Introduction and Section 3 already.    

  1. It’s not clear how does the machine learning approach is providing an added benefit for classifying hand gestures. In fact, the sensitivity analysis provided in section 3 shows a good agreement between the hand gesture and data. For instance, the forward and backward movements indicate that acceleration ax is the most represented and relevant as controlling data comparing to others acceleration’s directions.  I’m wondering for the sake of clarity why these data are not just used simply to control the drone.  

Thank you for your comments. As you pointed out, some gestures depend on specific component dominantly, such as forward and backward. But most of gestures have various motion components which cannot be classified simply to use only one component, and some advanced approach, such as machine learning is needed.

  1. Unless I’m mistaken, the system used to track the drone position (gold standard/reference) is not described in the paper. Figure 10 shows measurement of the position of the drone thanks to several trials. Please introduce the technical features of the tracking system.

Thank you for your suggestion. We extracted the coordinates of drone from virtual flight environment. The sample time was 0.02 second, and the x, y, and z points were calculated from 3-dimensional space. The gold line in Figure 9 and the guide line in Figure 10 were used to reference for the users. The technical feature of the tracking system was added briefly in Section 4.

  1. Widening the number of participants would improve significantly the soundness of the experiments and enables to identify the effect of the learning curve.    

We agree with your comments and the widening participant numbers will be considered in the future work for improvement of soundness and learning effect evidence. 

Please see the attachment of the revised manuscript. 

Reviewer 2 Report

The topic of the manuscript and the content are well organized and presented and the findings very interesting. Nevertheless, there are some problems that I must point out:

1. There is no background/related work section.

2. Even cheap consumer drones now include cameras allowing users to navigate using their smartphones. Therefore, flying a drone without using its camera (as studied in this paper), diminishes the problem importance significantly. The authors must address this issue in order to strengthen the importance of the problem they try to solve.

3. Some issues that are not as important:

(a) Were the users experienced at flying drones? If so, at what degree?

(b) Why did not all of the users participate in the last experiment with the actual drone, and there was only one user (one of the authors?)

Author Response

Statement of Changes

Submission number: sensors-2183113

Wearable Drone Controller: Machine Learning-Based Hand Gesture Recognition and Vibrotactile Feedback

Ji-Won Lee and Kee-Ho YU

Dear Reviewers,

Thank you for your review, and we really appreciate the effort of the review and the comments from you. Your advice was invaluable to improve the quality of the paper and provided insight to carry out the further work.

According to the comments of the review, the paper was revised carefully throughout the previously submitted manuscript. The added parts of the paper are highlighted in yellow color and the reply to your comments/suggestions are addressed below each comment, respectively.

List of Changes

  1. (p2) Added the description of the related work and limitations.
  2. (p2) Added the reason of IMU instead of vision.
  3. (p8) Added the experience of participants about drone control.
  4. (p8) Added the technical feature of the tracking system in the simulation.

Reviewer #2

The topic of the manuscript and the content are well organized and presented and the findings very interesting. Nevertheless, there are some problems that I must point out:

  1. There is no background/related work section.

Thank you for your suggestion. The background of this study is as follows: (1) Due to the lack of autonomous level, manual control by human operators is necessary. (2) Natural user interface for human operators is required to achieve good performance. (3) We proposed machine learning-based hand gesture control with IMU and vibrotactile feedback approach. These were stated in Introduction, for example, lines 27-31, lines 32-43, and lines 62-74 of page2, respectively. We added the description of the related work and limitations in Introduction.

  1. Even cheap consumer drones now include cameras allowing users to navigate using their smartphones. Therefore, flying a drone without using its camera (as studied in this paper), diminishes the problem importance significantly. The authors must address this issue in order to strengthen the importance of the problem they try to solve.

Thank you for your suggestion. The cameras are widely used in drones, but the vision is easily affected by light conditions and require tedious calibrations. On the other hand, the IMU has robust features in motion acquisition compared to conventional vision in general. This was stated at the end of second paragraph and added briefly in Introduction.

  1. Some issues that are not as important:

(a) Were the users experienced at flying drones? If so, at what degree?

Most participants have seldom experience at flying drones and the degree is beginner. We added briefly in Section 4.

(b) Why did not all of the users participate in the last experiment with the actual drone, and there was only one user (one of the authors?)

In the simulation experiment by the participants of 12 persons, the proposed interface showed good performance in control and perception of obstacles. The actual drone experiment was carried out by one of the participants to confirm the applicability of drone operation in real environment. 

Please see the attachment of the revised manuscript. 

Round 2

Reviewer 2 Report

The authors have performed all necessary actions.